# Dual-target peripheral and central magnetic stimulation for rehabilitation of chronic pelvic pain syndrome associated with psychosomatic symptoms: Study protocol for a randomized controlled trial

Chunmei Luo[1,2], Jiabei He[3], Degui Chang[4,5☯*], Haibo Lan[2], Meizhu Zhao[2], Xiaobin Zhen[2], Ren Liu[2], Lanjin Bai[2], Xueqian Li[2], Siyi Tian[2], Xiangdong Yang[2☯*]

1 Clinical Medical College, Chengdu University of Traditional Chinese Medicine, Chengdu, Sichuan Province, China, 2 Department of Anorectal, Chengdu Anorectal Hospital, Chengdu, Sichuan Province, China, 3 Department of Anorectal, Sichuan Province Fifth People's Hospital, Chengdu, Sichuan Province, China, 4 Department of Urology, Affiliated Hospital of Chengdu University of Traditional Chinese Medicine, Chengdu, Sichuan Province, China, 5 TCM Regulating Metabolic Diseases Key Laboratory of Sichuan Province, Affiliated Hospital of Chengdu University of Traditional Chinese Medicine, Chengdu, Sichuan Province, China

☯ These authors contributed equally to this work.
* 624440310@qq.com (DC); y-xd@vip.163.com (XY)

## Abstract

### Introduction

Chronic pelvic pain syndrome (CPPS) is frequently associated with psychological issues. Repetitive peripheral magnetic stimulation (rPMS) is potentially effective in treating CPPS, while repetitive transcranial magnetic stimulation (rTMS) has demonstrated therapeutic effects on anxiety and depression. Therefore, the study proposed herein aims to assess the efficacy and safety of dual-target magnetic stimulation in CPPS patients with psychological disorders.

### Methods

This prospective, double-blind, randomized controlled trial will recruit 75 CPPS participants. After stratification by sex, participants will be randomly assigned via block randomization (1:1:1), sequentially based on enrollment order, to one of three groups: dual-target magnetic stimulation (rPMS and rTMS), rPMS, and sham stimulation, all receiving standard treatment. The dual-site magnetic stimulation group will receive left dorsolateral prefrontal cortex (DLPFC) rTMS (120% resting motor threshold [RMT], 10 Hz, 4-s stimulation, 26-s interval, 3000 pulses in total [depression cases]) or right DLPFC rTMS (120% RMT, 1 Hz, 10-s stimulation, 2-s interval, 1000 pulses in total [anxiety cases]) combined with rPMS (50% of maximum stimulation intensity, 20 Hz, 2-s stimulation, 28-s interval, 1600 pulses in total). The rPMS group will receive

**Data availability statement:** No datasets were generated or analysed during the current study. All relevant data from this study will be made available upon study completion.

**Funding:** This project received support from the Tianfu Scientific Research Incubation Fund (NO. 2022QN02) and the Chengdu Medical Research Project (Grant No. 2024232). The funders had no role in study design, data collection and analysis, decision to publish, or preparation of the manuscript.

**Competing interests:** The authors have declared that no competing interests exist.

**Abbreviations:** CPPS: Chronic pelvic pain syndrome; EAU: European Association of Urology; FAS: Full analysis set; ITT: Intention-to-treat; PP: Per-protocol; QoL: Quality of life; RCT: Randomized controlled trial; RMT: Resting motor threshold.

only rPMS. The sham stimulation group will undergo sham transcranial and peripheral stimulation. All treatments will be administered five times a week, once daily, for 4 weeks. Primary outcomes will be the pelvic pain scale (females) or the National Institutes of Health Chronic Prostatitis Symptom Index (NIH-CPSI, males). Secondary outcomes will include assessment of pelvic floor muscle surface electromyography, pudendal nerve motor evoked potentials, the Depression, Anxiety, and Stress Scale (DASS-21), and the Short Form 36 (SF-36) quality of life scale.

## Discussion

We hypothesize that dual-target magnetic stimulation will show greater effectiveness than rPMS and sham stimulation in relieving pain symptoms and psychological distress in CPPS patients with comorbid mental disorders.

## Clinical trial registration

The study was prospectively registered at the Chinese Clinical Trial Registry (ChiCTR; http://www.chictr.org.cn, ID: ChiCTR2300078761) on December 18, 2023; Protocol version 1.0–20220709.

## Introduction

Chronic pelvic pain syndrome (CPPS) is characterized by persistent or recurrent pain in the pelvic area lasting >3 months in the absence of confirmed infection or other obvious local pathological causes [1]. Globally, CPPS has a high incidence, affecting 2%–16% of men and up to 24% of women [2,3]. CPPS can present with severe symptoms that may greatly impact psychological well-being causing anxiety, depression, and pain catastrophizing, placing increased pressure on families and society [4,5]. The precise causes and mechanisms of CPPS remain unclear, and the condition may be the end result of the interaction between multi-system dysfunction and psychological factors [6]. Therefore, addressing psychosocial factors in CPPS treatment is crucial [6–8].

Currently, the treatment of CPPS often involves physical therapy, medication management, and nerve blocks [9,10]. Studies have shown that repetitive peripheral magnetic stimulation (rPMS) may be effective in treating CPPS [11–13], while repetitive transcranial magnetic stimulation (rTMS) has a therapeutic effect on anxiety and depression [14–18]. However, investigations on PMS for treating CPPS are mostly observational studies or small sample studies, and there is currently a lack of high-quality research on the treatment of CPPS with PMS. Additionally, the presence of comorbid mental disorders has not been taken into consideration. Therefore, this trial protocol is designed to evaluate the efficacy and safety of combining rPMS with rTMS in CPPS patients with comorbid mental disorders, with the goal of providing a more effective treatment option for this patient population.

## Methods and analysis

### Trial design

Our study employs a sex-stratified, block-randomized controlled trial (RCT) design. After obtaining informed consent, eligible patients will first be stratified by sex. Within each stratum, participants will be assigned via block randomization (1:1:1), sequentially based on enrollment order, to one of the following three groups: (A) Dual-target magnetic stimulation (rPMS followed by rTMS); (B) Peripheral stimulation (rPMS); and (C) Sham stimulation. Recruitment will continue until 30 male and 45 female recruits have been registered. Assessment will be carried out prior to the initiation of treatment, at the end of each weekly treatment, and eight weeks after the entire treatment plan is finished. Figs 1 and 2 provides a detailed explanation of this program.

The objective of this study was to design a randomized, double-blind clinical trial to explore the efficacy and safety of dual-targeted magnetic stimulation for the treatment of CPPS associated with psychosomatic disorders. The study flow was performed in strict accordance with the CONSORT 2010-Flow Diagram (see Fig 3).

### Recruitment and study setting

Researchers will recruit participants via both online and offline channels, including the screening of outpatients and inpatients from the Chengdu Anorectal Hospital in Chengdu, China. This hospital is a tertiary Grade A specialized hospital, and its Proctology Department is a national key clinical specialty. The treatment procedures will be conducted at the Pelvic Floor Center of the hospital. A designated individual will assess and screen patients interested in participating in the study based on the inclusion and exclusion criteria to determine their eligibility. Inpatients will obtain permission from the attending physician, and those meeting the inclusion and exclusion criteria will sign a written informed consent form prior to the initiation of the treatment intervention.

### Participants

**Ethical approval and communication.** The research protocol for this RCT was approved by the Ethics Committee of Chengdu Proctology Hospital on October 30, 2023 (LL20231030). Any changes to the research protocol will be submitted for approval by this ethics committee. The pilot program will strictly adhere to the CONSORT statement. The RCT was registered on December 18, 2023 at www.ChiCTR.org, with registration number ChiCTR2200055615. After reviewing the information booklet about the project and its potential risks and benefits, the participants will be encouraged to discuss any questions with the researchers and will provide written informed consent to partake in this study. The clinical data collected will be kept by a dedicated person and will only be viewed by statistical analysts at the end of the RCT. The RCT results will be published in a peer-reviewed journal according to the standards of the International Committee of Medical Journal Editors and will not involve professional authors. Undisclosed trial data will be stored in a public database at the time of publication. The RCT results will also be disseminated through scientific conferences. Consent for publication: Written informed consent for publication was obtained from all participants.

**Study population.** The study population will include 30 males and 45 females between the ages of 18–70 years, experiencing CPPS accompanied by anxiety or depression.

**Diagnostic criteria.**

1. According to the European Association of Urology (EAU), study participants will be required to meet the CPPS diagnostic criteria outlined in the 2022 edition of the Chronic Pelvic Pain Guidelines: (1) pelvic pain in the relevant area lasting >3 months, without evidence of an identifiable infection or local disease causing the pain; (2) concurrent lower urinary tract symptoms in the anterior pelvic region, dysfunction of the central pelvic floor, and organ dysfunction, in addition to dysfunction of the posterior pelvic floor, involving the rectum and anus.

# Table 1　SPIRIT schedule

| Research Phase / Project Name | Before Treatment t0 | First Week Treatment t1(5days) | Second Week Treatment t2(5days) | Third Week Treatment t3(5days) | Fourth Week Treatment t4(5days) | 8 Weeks After Treatment t5 |
|---|---|---|---|---|---|---|
| Screening for Enrollment | √ | | | | | |
| Medical history and basic information | √ | | | | | |
| Routine examinations | √ | | | | √ | |
| Informed Consent Form | √ | | | | | |
| Drug treatment | √ | √ | √ | √ | √ | √ |
| Pelvic Pain Scale/NIH-CPSI Score | √ | √ | √ | √ | √ | √ |
| DASS-21Score | √ | √ | √ | √ | √ | √ |
| SF-36 Quality of Life Scale | √ | √ | √ | √ | √ | √ |
| Surface electromyography of the pelvic floor muscles | √ | √ | √ | √ | √ | √ |
| Pudendal nerve motor evoked potentials | √ | √ | √ | √ | √ | √ |
| Bilateral magnetic stimulation | | √ | √ | √ | √ | |
| Peripheral magnetic stimulation | | √ | √ | √ | √ | |
| Fake stimulus | | √ | √ | √ | √ | |
| Follow-up | | | | | | √ |
| Adverse Event | | √ | √ | √ | √ | √ |
| Analysis of missing items and non-respondents. | | | | | | √ |

Note: Routine examinations include: complete blood count, liver function, kidney function, and electrocardiogram examination. Pelvic pain scale, Female Pelvic Pain Scoring Scale; NIH-CPSI, The National Institutes of Health Chronic Prostatitis Symptom Index (NIH-CPSI); SF-36,SF-36 scale of quality of life; t0, enrollment; t1, At the end of 5 sessions;t2,At the end of 10 sessions; t3,At the end of 15 sessions; t4,At the end of 20 sessions; t5, Follow-up visit 8 weeks after treatment ends.

**Fig 1. SPIRIT schedule.**

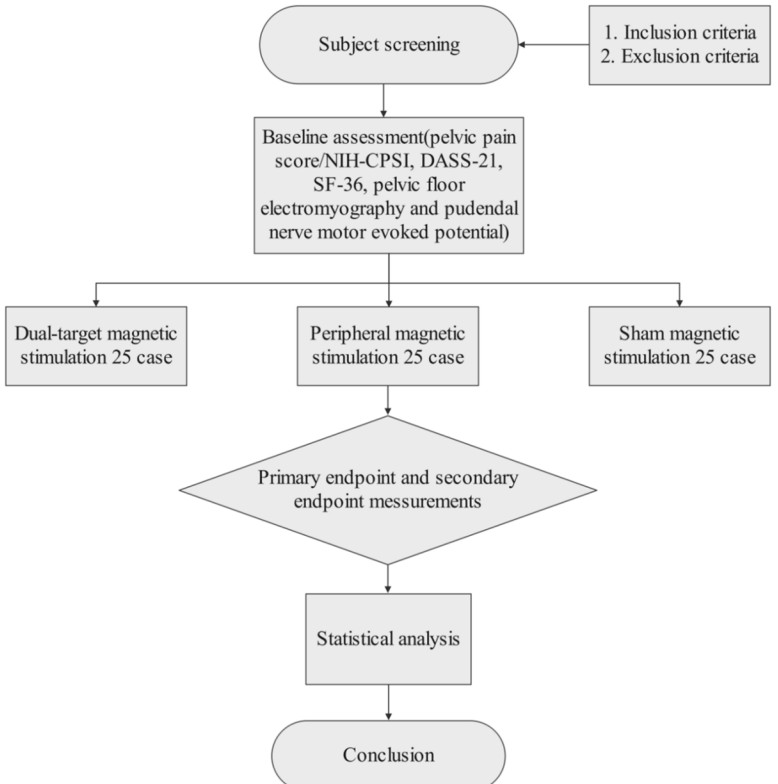

**Fig 2. Clinical research flowchart.**

2. Diagnosis criteria for generalized anxiety disorder, according to the Diagnostic and Statistical Manual of Mental Disorders, 5th Edition (DSM-5-TR) and International Classification of Diseases 11th Revision (ICD-11) standards: (1) People experience excessive anxiety and worry about many daily events (concerning mainly family, health, financial situation, school, and work) for at least 6 months; (2) difficulty controlling this worry; (3) the anxiety and worry are accompanied by at least 3 of the following symptoms: restlessness or feeling on edge, easily fatigued, difficulty concentrating or mind going blank, irritability, muscle tension, sleep disturbances, motor fidgeting, and sympathetic overactivity; (4) the anxiety, worry, or physical symptoms cause clinically significant distress or impairment in social, occupational, or other important areas of functioning; (5) the disorder is not attributable to a substance; (6) the disorder is not better explained by another mental disorder.

3. Diagnosis criteria for major depressive disorder according to DSM-5-TR and ICD-11 standards: (1) at least 5 of the following symptoms are present within 2 weeks: depressed mood almost every day for most of the day, markedly diminished interest or pleasure in activities almost every day for most of the day, decrease or increase in appetite almost every day, or significant weight loss or gain (without dieting), insomnia or hypersomnia almost every day, psychomotor agitation or retardation almost every day, fatigue or loss of energy almost every day, feelings of worthlessness or excessive guilt almost every day, diminished ability to think or concentrate, indecisiveness almost every day, recurrent thoughts of death, significant reduction in energy or vigor, thoughts of self-harm, and being in a state of extreme anger or irritability; (2) these symptoms cause clinically significant distress or impairment in social, occupational, or other important areas of functioning; (3) the symptoms are not attributable to the physiological effects of a substance or another medical condition; (4) the symptoms are not better explained by schizoaffective disorder, schizophrenia,

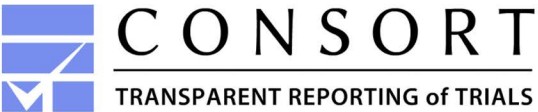

## CONSORT 2010 Flow Diagram

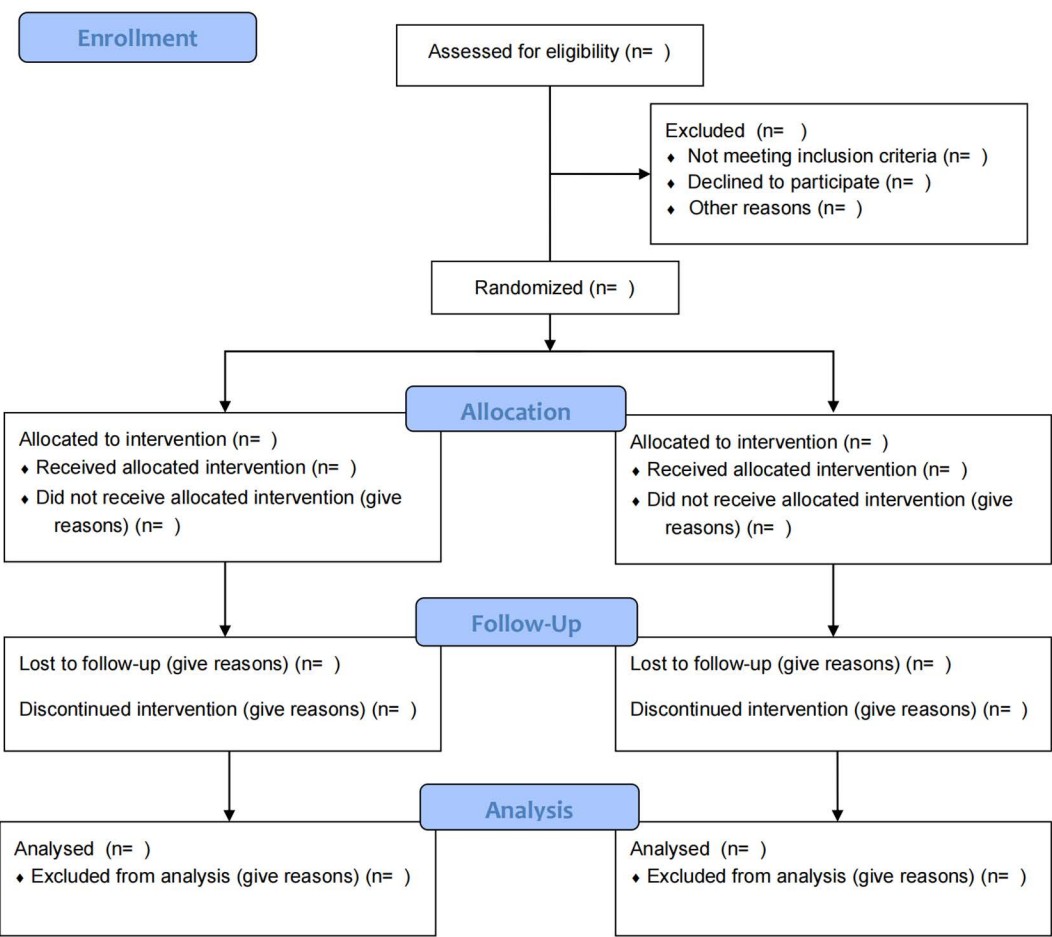

**Fig 3. CONSORT-2010-flow-diagram.**

schizophreniform disorder, delusional disorder, or other specified or unspecified schizophrenia spectrum and other psychotic disorders; (5) there has never been a manic or hypomanic episode.

**Inclusion criteria.**

1. Meeting the CPPS diagnostic criteria in the 2022 EAU guidelines.

2. Individuals aged between 18 and 70 years.

3. Definitive generalized anxiety disorder or major depressive disorder.

4. No identifiable pathological changes in physical examinations and auxiliary tests.

5. No treatment other than oral medications in the 3 months prior to the visit.

6. Patient's informed consent and voluntary participation in the study.

**Exclusion criteria.**

1. Patients in the acute phase of systemic and intracranial hemorrhagic diseases.

2. Individuals with serious underlying conditions, such as cardiovascular, liver, kidney, respiratory, and blood disorders, in addition to malignant tumors and other advancing illnesses.

3. Patients with cardiac metal valves, cardiac pacemakers, intracranial metal implants, lumbar sacral metal implants, and implantable electronic devices.

4. Individuals with infections in the head or lumbar sacral regions.

5. Individuals exhibiting unstable vital signs.

6. Patients with previous adverse reactions to magnetic therapy.

7. Individuals with atypical autonomic reflexes.

8. Patients with cognitive impairment who cannot cooperate.

9. Expectant or breastfeeding women.

10. Patients with a history of diseases causing peripheral nerve damage.

11. Patients with debilitating diseases such as malignant effusion, active pulmonary tuberculosis, cancer, or myasthenia gravis.

12. Patients with severe mental illness or epilepsy.

**Sample size.** The sample size for each group was determined based on two previous clinical studies [11,19] using G*Power 3.1.9.7 software. The reference study reported an effectiveness rate of 54.5% for the treatment of pelvic floor discomfort associated with premenstrual syndrome (PMS) [11]. Additionally, significant results for transcranial magnetic stimulation in treating depression and anxiety were reported at 53.2% and 56.9%, respectively. The expected effect size was set at 0.5, with $\beta = 0.1$ and $\alpha = 0.05$ [11,19]. The sample size was increased by 10% due to the unknown distribution of the outcome data, as a nonparametric test was planned for the statistical analysis. Furthermore, the sample size was augmented by an additional 10% to account for a projected dropout rate of 10%.

According to epidemiological data, the male-to-female prevalence ratio is 16% to 24% (2:3) [2,3]. To ensure balanced grouping, it was determined that a minimum of 30 male participants was required, resulting in 45 female participants. Participants were assigned to three groups (A, B, and C), each consisting of 10 males and 15 females. Consequently, there were 25 participants in each group.

**Withdrawal criteria and management.** Participants will be required to withdraw from the RCT under the following circumstances: (1) if the patient does not adhere to the prescribed treatment plan or receives alternative treatment; (2) if the patient experiences severe adverse reactions or changes in their condition that prevent them from continuing in the trial; (3) if the patient experiences significant organ dysfunction, unstable blood pressure, heart rate, or breathing; (4) if the patient requests to withdraw their informed consent due to intolerable adverse reactions or for no specific reason; and (5) if the patient is lost to follow-up during the treatment or follow-up period.

**Stratified randomization.** The enrolled patients were assigned numbers based on gender: women were numbered from 101 to 145, while men were numbered from 201 to 230. Males and females were organized into one unit group for every three individuals in chronological order. The random number generator in SPSS 27.0 software was utilized to establish the random seed. The fixed starting point was set at 1000, with the maximum and minimum values calibrated between 1000 and 1100 using the 'Rv.Uniform' function within the 'Random Number' function group to generate random numbers. Within each unit group, the random numbers were sorted by size, with the smallest value assigned to group A (dual-target magnetic stimulation group), the middle value to group B (rPMS group), and the largest value to group C (sham stimulation group).

## Blinding

An investigator will use sealed, opaque envelopes to conceal the order of allocation. After obtaining consent from the patient and confirming eligibility, the envelope corresponding to the patient's enrollment number will be handed over to the trial operator. The operator and assessor of the trial will be different investigators; the operator will be aware of the grouping but will not be involved in the assessment; the assessor will not be aware of the grouping and will only examine and assess the enrolled patients. After data collection is complete, a first unblinding will take place, during which the two investigators will enter and validate the data individually. Finally, statistical analyses will be performed by two statistical analysts who will compare the three groups without knowledge of the subgroups. Once the analyses are complete, a second unblinding will take place, marking the end of the study.

In the event of medical necessity or an emergency requiring unblinding, the final decision will be jointly made by the two study monitors after mutual consultation.

## Intervention

Clinical intervention measures and steps. 1) The operator of the trial conducts a risk assessment of the treatment after confirming the patient's information and treatment plan. 2) Stimulation planning: initiate with rPMS, followed by rTMS. 3) Stimulation sites: for rPMS, the center of the circular coil will be aligned with the third sacral segment. During stimulation, correct coil positioning will be confirmed if there is a sensation of contraction in the perianal muscles and a dorsiflexion response. For central stimulation, the left dorsolateral prefrontal cortex (l-DLPFC) area will be targeted in patients with depression, and the right DLPFC (r-DLPFC) area will be targeted in patients with anxiety [20]. 4) Stimulation intensity: for peripheral stimulation, 50% of maximum stimulation intensity will be applied. For central stimulation, the setting will be adjusted to 120% of the resting motor threshold [RMT] [21]. 5) Stimulation parameters: (1) peripheral stimulation: 20 Hz, 2-s stimulation, 28-s interval, a total of 1600 pulses; (2) r-DLPFC: 1 Hz, 10-s stimulation, 2-s interval, a total of 1000 pulses; (3) l-DLPFC: 10 Hz, 4-s stimulation, 26-s interval, a total of 3000 pulses. 6) End of stimulation: the stimulation coil will be moved away from the patient's stimulation site. The patient will be informed of the treatment's completion and the assessor will inquire about any discomfort. If the patient is lying down, they will be instructed to gradually sit up and then stand to avoid falling. 7) Treatment course: once a day, 5 days a week, for a total of 4 weeks (20 sessions). Treatment concludes after the 20th session.

**Dual-target magnetic stimulation procedure.** During treatment: determine the stimulation plan, and initiate with rPMS before rTMS. Adjust the seat to a horizontal angle of 10°–20°, instruct the patient to lie prone on the seat, align the center of the circular coil with the third sacral area, and stimulate until there is a sensation of contraction in the anal sphincter muscles and a dorsiflexion response, indicating correct device positioning. Use single-pulse stimulation to determine the stimulation intensity and repeat PMS according to the plan. Adjust the seat, have the patient sit upright, and measure the RMT during the initial treatment to determine the magnetic stimulation intensity. rTMS stimulation intensity is set at 120% of the RMT.

Auxiliary positioning device: Determine the position of the stimulated cortex based on the positioning cap; for depressed patients, stimulate the l-DLPFC, and for anxious patients, stimulate the r-DLPFC, following the central stimulation plan.

**Peripheral magnetic stimulation procedure.** Initially, administer rPMS following the same procedure as for the dual-target magnetic stimulation group. Sham rTMS will be subsequently applied using a coil that emits sound but does not produce a magnetic field, with parameter settings identical to those of rTMS in the dual-target magnetic stimulation group.

**Sham stimulation procedure.** Sham rPMS and rTMS stimulation will be conducted using appropriate coils that emit sound but do not produce a magnetic field. The remaining parameters and procedures will be the same as for the dual-target magnetic stimulation group.

**Post-treatment procedures.** A researcher serves as the evaluator of the patient's condition, assessing it through questions at the end of every five treatment sessions, promptly and completely filling out various records, and storing them for future reference. After each session, the treatment room and equipment will undergo a thorough disinfection process.

**Treatment regimen.** A treatment course will comprise 20 sessions, administered five times per week, with one treatment per day. The 20th treatment will mark the end of the course. Follow-up visits will be scheduled at 8 weeks after the end of the treatment. During the study period, patients may receive oral medication treatment, and the specific drugs and dosages will be recorded at the time of enrollment. If there are any significant adjustments to the type and dosage of the patient's medication, they will need to inform the therapist proactively.

**Emergency situation handling.** The therapeutic equipment must incorporate protective measures, such as emergency braking devices, as required by GB9706.1−2020 medical device safety standards, such as emergency braking devices. During the treatment process, if there is severe damage to vital organ function, loss of consciousness, or other emergency situations, immediate handling and first aid should be provided as follows: a) immediately stop treatment; b) ensure the patient's airway is clear; c) carefully monitor the patient's vital signs; d) in the case of cardiac or respiratory arrest, promptly administer cardiopulmonary resuscitation; and e) contact the patient's family to inform them of the situation and appropriate measures. After resolution of the emergency situation, summarize the experience and lessons learned.

**Outcome assessment**

**Baseline indicators.**

1. Age, blood pressure, body temperature, pulse, complete medical history, comorbidities, and concurrent medication data.

2. Pelvic floor surface electromyography and results of motor evoked potentials from the pudendal nerve.

3. Pelvic pain score/NIH-CPSI, DASS-21, and SF-36 score results.

4. Baseline results for the electrocardiogram, complete blood count, and urinalysis.

**Observation items.** Pelvic pain score/NIH-CPSI, DASS-21 scale, SF-36 score scale, pelvic floor surface electromyography, and results of pudendal nerve motor evoked potentials.

1. Main indicators: Pain score

(1) Pelvic pain szcore sheet (for females): includes the patient's basic information, detailed symptom changes, and the effectiveness of previous treatments. It uses exacerbation and relief of pain associated with sexual activity and menstruation as the starting point for medical history inquiry. Supplementary details include symptoms related to exercise and urinary and gastrointestinal function. Please refer to Table 1 for further details.

**Table 1. Pelvic pain score sheet.**

| Number: | | | | | Name: | | | | | |
|---|---|---|---|---|---|---|---|---|---|---|
| ID: | | | | | Tel: | | | | | |

Pain characteristics

1, Please describe your pain situation, such as frequency, duration, and intensity of the pain.

2, What do you think is the cause of the pain?

3, Has there been any event related to the onset of pain? A.Yes B.No

If so, what is the related event?

4, How long has this been going on?

5, The following symptoms are graded based on the severity of the past week. Please √ the appropriate box.

| Symptoms | "Zero to Ten"Scale (0–10)for rating pain | | | | | | | | | | |
|---|---|---|---|---|---|---|---|---|---|---|---|
| | 0 | 1 | 2 | 3 | 4 | 5 | 6 | 7 | 8 | 9 | 10 |
| | NO Pain | | | | | Moderate Pain | | | | | Worst Pain |
| | 0 | 1 | 2 | 3 | 4 | 5 | 6 | 7 | 8 | 9 | 10 |
| How do you evaluate pain? | | | | | | | | | | | |
| Ovulation pain (mid-menstrual cycle) | | | | | | | | | | | |
| Premenstrual abdominal pain | | | | | | | | | | | |
| Premenstrual pain (no spasms) | | | | | | | | | | | |
| Deep dyspareunia | | | | | | | | | | | |
| Groin pain when lifting leg | | | | | | | | | | | |
| Post-coital pain lasting hours or days | | | | | | | | | | | |
| Pain when holding urine | | | | | | | | | | | |
| Muscle/joint pain | | | | | | | | | | | |
| Menstrual cramping pain | | | | | | | | | | | |
| Pain disappears after menses | | | | | | | | | | | |
| Burning pain in the vagina after sex | | | | | | | | | | | |
| painful urination | | | | | | | | | | | |
| backache | | | | | | | | | | | |
| migraines | | | | | | | | | | | |
| seating pain | | | | | | | | | | | |
| Overall score | | | | | | | | | | | |

(2) NIH-CPSI (for males): mainly consists of three parts, which assess the pain or discomfort caused by chronic prostatitis, urinary symptoms, and the impact on quality of life (QoL), with a total of nine questions. It has the characteristics of being objective and convenient, is quickly accepted by patients, and can provide important references for scientific research and clinical work. Please refer to Table 2 for further details.

2. Secondary indicators:

(1) Anxiety and depression score (DASS-21): regarding depression, critical values for mild, moderate, and severe depression are 10, 14, and 21, respectively; for the anxiety scale, critical values for mild, moderate, and severe anxiety are 8, 10, and 15, respectively; for the stress scale, critical values for mild, moderate, and severe stress are 15, 19, and 26, respectively. Please refer to Table 3 for further details.

(2) Surface electromyography of the pelvic floor muscles: the participant will lie in the supine position with their upper and lower body forming an angle of approximately 120°, with their feet naturally rotated outwards. A rectal electrode will be placed to collect surface electromyography data of the pelvic floor muscles, while abdominal electrodes will be used to monitor abdominal muscle activity. Before the examination, the participant will be instructed to urinate

**Table 2. NIH-CPSI.**

| Number: | | | Name: | | | | |
|---|---|---|---|---|---|---|---|
| ID: | | | Tel: | | | | |

The following symptoms are graded based on the severity of the past week. Please √ the appropriate box.

| Pain or discomfort symptoms | | None | Rarely | Occasionally | Frequently | Very common | Almost always |
|---|---|---|---|---|---|---|---|
| **1.**Have you experienced pain or discomfort in the following areas? | | 0 | 1 | 2 | 3 | 4 | 5 |
| Perineal pain | | | | | | | |
| Didymalgia | | | | | | | |
| Glans penis | | | | | | | |
| Lumbosacral and suprapubic region | | | | | | | |
| **2.**Pain or burning sensation during urination | | | | | | | |
| **3.**Painful discomfort at or after ejaculation | | | | | | | |

4.Use numbers to describe the degree of pain or discomfort mentioned above.
No pain 0 1 2 3 4 5 6 7 8 9 10 worst possible pain

| Urinary symptoms | | None | Less than 1/5 | Less than half | About half | More than half | Almost every time |
|---|---|---|---|---|---|---|---|
| | | | 1 | 2 | 3 | 4 | 5 |
| 5.Whether there is often a feeling of incomplete urination at the end of urination? | | | | | | | |
| 6.Do you often feel the need to urinate again within 2 hours after urinating? | | | | | | | |

| Symptom severity | 0 | None | Little | Some | A lot | | |
|---|---|---|---|---|---|---|---|
| | | | 1 | 2 | 3 | | |
| 7.Do the above symptoms affect your daily life? | | | | | | | |
| 8.Are you always reminded of your symptoms? | 0 | | | | | | |

| Quality of life | Very satisfactory | Satisfactory | Good | Just so so | Mostly unsatisfactory | Unhappy | Fearful |
|---|---|---|---|---|---|---|---|
| | 0 | 1 | 2 | 3 | 4 | 5 | 6 |
| If left untreated, how do you think your life will be in the future? | | | | | | | |

Analysis of NIH-CPSI score results

| Pain and discomfort score:1+2+3+4=( ) | | Symptom impact scores on quality of life: 7+8+9=( ) | | |
|---|---|---|---|---|
| Voiding Symptom Score: 5+6=( ) | | | | |
| Symptom severity: 1+2+3+4+5+6=( ) | | Mild:0~9 | Moderate:10~18 | Severe:19~31 |
| Overall rating: 1+2+3+4+5+6+7+8+9=( ) | | Mild:1~14 | Moderate:15~29 | Severe:30~43 |

and defecate, taught how to correctly contract and relax the pelvic floor, and will be informed about the assessment process. They will learn how to quickly contract and maintain a 10-s contraction, receiving guidance to contract and relax the pelvic floor muscles according to voice prompts to record surface electromyography values.

(3) Pudendal nerve motor evoked potentials: participants will be instructed to use glycerin suppositories for rectal emptying and assume a prone position. The magnetic stimulator coil will be placed 3–5 cm to the side of the midline of the S3 plane, stimulating both sides at an intensity of 55%–60% of the maximum output. A surface electrode will be placed in the rectum to record the anal sphincter, and the ground will be connected to the wrist. The amplifier has a sampling bandwidth of 5–2000 Hz, sensitivity of 3 ms per division, and an analysis time of 30 ms. Five successful recordings will be made, and the average result will be taken.

**Table 3. Depression anxiety and stress scale(DASS-21).**

| Number: | | Name: | | | | |
|---|---|---|---|---|---|---|
| ID: | | Tel: | | | | |

(The subscale scores were multiplied by 2 to give a score for that subscale, with higher scores representing more of that emotion.)

Please read each statement and indicate the extent to which it applies to you in the past week by checking the appropriate box. There are no right or wrong answers. Do not spend too much time on any statement.The scoring criteria are as follows:
0 Does not apply to me at all
1 Applies to me to some extent, or applies sometimes
2 Applies to me to a onsiderable extent, or applies most of the time
3 Applies to me very much, or applies most of the time

| | | 0 | 1 | 2 | 3 | score |
|---|---|---|---|---|---|---|
| 1 | I couldn't seem to experience any positive feeling at all | | | | | |
| 2 | I found it difficult to work up the initiative to do things | | | | | |
| 3 | I felt that I had nothing to look forward to | | | | | |
| 4 | I felt down-hearted and blue | | | | | |
| 5 | I was unable to become enthusiastic about anything | | | | | |
| 6 | I felt I wasn't worth much as a person | | | | | |
| 7 | I felt that life was meaningless | | | | | |
| Depression Scale | | | | Total score | | |
| 8 | I was aware of dryness of my mouth | | | | | |
| 9 | I experienced breathing difficulty (e.g., excessively rapid breathing, breathlessness in the absence of physical exertion) | | | | | |
| 10 | I experienced trembling (e.g., in the hands) | | | | | |
| 11 | I was worried about situations in which I might panic and make a fool of myself | | | | | |
| 12 | I felt I was close to panic | | | | | |
| 13 | I was aware of the action of my heart in the absence of physical exertion (e.g., sense of heart rate increase, heart missing a beat) | | | | | |
| 14 | I felt scared without any good reason | | | | | |
| Anxiety Scale | | | | Total score | | |
| 15 | I found it difficult to relax | | | | | |
| 16 | I tended to over-react to situations | | | | | |
| 17 | I felt that I was using a lot of nervous energy | | | | | |
| 18 | I felt that I was rather touchy | | | | | |
| 19 | I found it hard to wind down | | | | | |
| 20 | I was intolerant of anything that kept me from getting on with what I was doing | | | | | |
| 21 | I found myself getting agitated | | | | | |
| Stress Scale | | | | Total score | | |

Recommended cut-off scores for conventional severity labels (normal, moderate, severe) are as follows: NB Scores on the DASS-21 will need to be multiplied by 2 to calculate the final score.

| | Normal | Mild | Moderate | Severe | Extremely Severe |
|---|---|---|---|---|---|
| Depression | 0-9 | 10-13 | 14-20 | 21-27 | 28+ |
| Anxiety | 0-7 | 8-9 | 10-14 | 15-19 | 20+ |
| Stress | 0-14 | 15-18 | 19-25 | 26-33 | 34+ |

Lovibond, S.H. & Lovibond, P.F. (1995). Manual for the Depression Anxiety & Stress Scales. (2nd Ed.)Sydney: Psychology Foundation.

(4) SF-36 QoL scale: a universal quantitative scale consisting of 36 items, including physical functioning, physical role, bodily pain, general health, vitality, social functioning, emotional role, and mental health, across eight domains. Please refer to Table 4 for further details.

## Safety and participant compensation

Prior to initiating TMS, participants will be provided with hearing protection. During the intervention, adverse event monitoring will be conducted for participants. Participants will be asked about adverse events after each stimulation session until the events are resolved. If headaches occur after treatment, relaxation therapy will be provided. In the case of falls or other incidents during treatment, participants will receive free treatment from a professional therapist until recovery. Participants identified as at risk of suicide, self-neglect, neglect of others, or experiencing serious adverse events will be referred to relevant clinical services. The causality and severity of adverse events related to rTMS will be evaluated. Serious adverse events will be reported to the Chengdu Proctology Hospital Ethics Committee within 72 hours. Participants retain the right to pursue compensation in the event of any unexpected incidents associated with rPMS or rTMS. Any adverse events that result in participant withdrawal will be documented and reported.

## Statistical analyses

Statistical analyses will be carried out using SPSS 27.0 software, with the normality of the variables being assessed using the Shapiro-Wilk test. In the event of continuous variables demonstrating a normal distribution, subsequent reporting will be based on mean±SD and 95% CI. Conversely, if continuous variables are found not to be normally distributed, they will be reported using medians and interquartile ranges. Categorical variables are expressed as absolute frequencies and percentages.

Repeated measures ANOVA will be used for group comparisons, followed by post hoc tests for multiple comparisons. For paired sampling, paired t-test for parametric data and Wilcoxon test for non-parametric data are used to compare pre- and post-intervention outcomes within groups, as necessary based on the data distribution. We will quantify effect sizes by calculating Cohen's d. We will adjust for confounding variables for regression analyses. Statistical significance will be indicated by a P value < 0.05. In case of participant withdrawal during the trial, an intention-to-treat (ITT) analysis will be conducted, comparing the full analysis set (FAS) and per-protocol set (PP).

## Collection and management of data

All clinical data will be entered by two dedicated data collectors for unified data entry and verification. The data will then be extracted by data managers and handed over for statistical analysis to researchers who are blinded to the grouping. The final unblinding will take place only after the analysis is complete. The analysis results will be published through articles, and all data will be reported in the results database. In addition, missing data (e.g., loss to follow-up, death, or withdrawal) will be updated using multiple imputation methods.

## Oversight and monitoring

The double-blind management system will be strictly followed, and the supervisory committee, comprising relevant members of the hospital's ethics committee, will oversee data safety and the blinding of the experimental process.

## Patient and public involvement

Before finalizing the study protocol, individual patients and experienced therapists (unrelated to the included cases) will be invited to assess the feasibility of the RCT design and the planned interventions. Based on their feedback, adjustments to assessment methods and treatment timing will be made to avoid disrupting regular treatment schedules. Patients or public

**Table 4. 36-item quality of life scale.**

| Number: | | | | | Name: | | |
|---|---|---|---|---|---|---|---|
| ID: | | | | | Tel: | | |

There are a total of 36 questions below, and after each question, there are several answers to choose from. Please fill in the corresponding score in the score column of the table for the answers you think are appropriate.

| Item | | | | | | | Score |
|---|---|---|---|---|---|---|---|
| 1. In general, would you say your health is | Excellent | Very good | Good | | Fair | Poor | |
| | 5 | 4.4 | 3.4 | | 2 | 1 | |
| 2. **Compared to one year ago**, how would you rate your health in genera **now**? | Much better | Somewhat better | About the same | | Somewhat worse | Much worse | |
| | 1 | 2 | 3 | | 4 | 5 | |
| The following items are about activities you might do during a typical day. Does **your health now limit you** in these activities? If so, how much? | | | | Yes, limited a lot | Yes, limited a little | No, not limited at all | |
| 3. **Vigorous activities**, such as running, lifting heavy objects, participating in strenuous sports | | | | 1 | 2 | 3 | |
| 4. **Moderate activities**, such as moving a table, pushing a vacuum cleaner, bowling, or playing golf | | | | 1 | 2 | 3 | |
| 5. Lifting or carrying groceries | | | | 1 | 2 | 3 | |
| 6. Climbing **several** flights of stairs | | | | 1 | 2 | 3 | |
| 7. Climbing **one** flight of stairs | | | | 1 | 2 | 3 | |
| 8. Bending, kneeling, or stooping | | | | 1 | 2 | 3 | |
| 9. Walking **more than a mile** | | | | 1 | 2 | 3 | |
| 10. Walking **several blocks** | | | | 1 | 2 | 3 | |
| 11. Walking **one block** | | | | 1 | 2 | 3 | |
| 12. Bathing or dressing yourself | | | | 1 | 2 | 3 | |
| During the **past 4 weeks**, have you had any of the following problems with your work or other regular daily activities **as a result of your physical health**? | | | | No | Yes | | |
| 13. Cut down the **amount of time** you spent on work or other activities | | | | 1 | 2 | | |
| 14. **Accomplished less** than you would like | | | | 1 | 2 | | |
| 15. Were limited in the **kind** of work or other activities | | | | 1 | 2 | | |
| 16. Had **difficulty** performing the work or other activities (for example, it took extra effort) | | | | 1 | 2 | | |
| During the **past 4 weeks**, have you had any of the following problems with your work or other regular daily activities **as a result of any emotional problems** (such as feeling depressed or anxious)? | | | | No | Yes | | |
| 17. Cut down the **amount of time** you spent on work or other activities | | | | 1 | 2 | | |
| 18. **Accomplished less** than you would like | | | | 1 | 2 | | |
| 19. Didn't do work or other activities as **carefully** as usual | | | | 1 | 2 | | |
| 20. During the **past 4 weeks**, to what extent has your physical health or emotional problems interfered with your normal social activities with family, friends, neighbors, or groups? | Not at all | Slightly | Moderately | Quite a bit | Extremely | | |
| | 5 | 4 | 3 | 2 | 1 | | |
| 21.How much **bodily** pain have you had during the **past 4 weeks**? | None | Very mild | Mild | Moderate | Severe | Very severe | |
| | 6 | 5.4 | 4.2 | 3.1 | 2.2 | 1 | |
| 22.During the past 4 weeks, how much did pain interfere with your normal work (including both work outside the home and housework)? | Score selection | Not at all | A little bit | Moderately | Quite a bit | Extremely | |
| | Answered items 21 | 6 | 4.75 | 3.5 | 2.25 | 1 | |
| | Unanswered items 21 | 5 | 4 | 3 | 2 | 1 | |

*(Continued)*

**Table 4.** (Continued)

| These questions are about how you feel and how things have been with you during the past 4 weeks. For each question, please give the one answer that comes closest to the way you have been feeling. How much of the time during the past 4 weeks... | All of the time | Most of the time | A good bit of the time | Some of the time | A little of the time | None of the time | | |
|---|---|---|---|---|---|---|---|---|
| 23.Did you feel full of pep? | 6 | 5 | 4 | 3 | 2 | 1 | | |
| 24.Have you been a very nervous person? | 1 | 2 | 3 | 4 | 5 | 6 | | |
| 25.Have you felt so down in the dumps that nothing could cheer you up? | 1 | 2 | 3 | 4 | 5 | 6 | | |
| 26.Have you felt calm and peaceful? | 6 | 5 | 4 | 3 | 2 | 1 | | |
| 27.Did you have a lot of energy? | 6 | 5 | 4 | 3 | 2 | 1 | | |
| 28.Have you felt down hearted and blue? | 1 | 2 | 3 | 4 | 5 | 6 | | |
| 29.Did you feel worn out? | 1 | 2 | 3 | 4 | 5 | 6 | | |
| 30.Have you been a happy person? | 6 | 5 | 4 | 3 | 2 | 1 | | |
| 31.Did you feel tired? | 1 | 2 | 3 | 4 | 5 | 6 | | |
| 32.During the **past 4 weeks**, how much of the time has **your physical health or emotional problems** interfered with your social activities (like visiting with friends, relatives, etc.)? | All of the time | Most of the time | Some of the time | A little of the time | None of the time | | | |
| | 1 | 2 | 3 | 4 | 5 | | | |
| How TRUE or FALSE is **each** of the following statements for you. | Definitely true | Mostly true | Don't know | Mostly false | Definitely false | | | |
| 33. I seem to get sick a little easier than other people | 1 | 2 | 3 | 4 | 5 | | | |
| 34.I am as healthy as anybody I know | 5 | 4 | 3 | 2 | 1 | | | |
| 35. I expect my health to get worse | 1 | 2 | 3 | 4 | 5 | | | |
| 36. My health is excellent | 5 | 4 | 3 | 2 | 1 | | | |

| Score conversion = (actual score – lowest score in that aspect)/ (highest score in that aspect – lowest score in that aspect) × 100; missing entry scores are replaced by the average score of their respective aspect. | conversion formula | Score conversion |
|---|---|---|
| Physical Functioning(PF), Consist of items 3–12, | PF=(actual score-10)/20 × 100 | |
| Role-Physical(RP),Consist of items 13–16, | RP=(actual score-4)/4 × 100 | |
| Bodily Pain(BP), Consist of items 21, 22, | BP=(actual score-2)/10 × 100 | |
| General Health(GH), Consist of items 1, 33–36, | GH=(actual score-5)/20 × 100 | |
| Vitality(VT), Consist of items 23, 27, 29, 31, | VT=(actual score-4)/20 × 100 | |
| Social Functioning(SF), Consist of items 20, 32, | SF=(actual score-2)/9 × 100 | |
| Role-Emotional(RE), Consist of items 17–19, | RE=(actual score-3)/3 × 100 | |
| Mental Health(MH), Consist of items 24–26, 28, 30, | MH=(actual score-5)/25 × 10 | |
| Reported Health Transition(HT), Consist of items 2, | HT=(actual score-1)/4 × 100 | |

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

members will not be involved in recruitment, data implementation, or measurements. Results will be communicated to participants through written reports. The costs of adverse reactions to the interventions included in this RCT will not be borne by the patients. During the intervention, participants will be under the care of the hospital's family physicians, ensuring that all medical records are up-to-date and that patients attend regular follow-ups. Interruptions in participants' interventions or data from the intervention plan will be recorded, to ensure adequate analysis of treatment effectiveness and safety outcomes.

## Discussion

CPPS is characterized by a multifaceted origin. It commonly encompasses heightened tension in the pelvic floor muscles, peripheral inflammation, peripheral and central sensitization, and is frequently influenced by psychosocial factors [22]. A series of studies have indicated that CPPS is closely related to psychosocial factors and often co-occurs with anxiety, depression, and pain catastrophizing cognition [15,23–26]. Some studies have suggested an association between chronic pain and reduced gray matter volume in the DLPFC [27]. Clinical practice has shown that the treatment effect of medications alone or combined with local nerve block is often not ideal. Currently, neuroregulation is increasingly used in the treatment of CPPS [1,28]. Non-invasive electrical and magnetic stimulation techniques are the most widely used neuroregulation techniques [29]. Purported effects include modulation of neural plasticity, neurotransmitter and growth factor release and balance, reconstruction of neural circuits, and proliferation of neural cells [30,31]. rPMS applied to peripheral muscles and nerves induces proprioceptive inputs to a target area in the central nervous system. It creates an electric field in deep tissue and directs electrical currents into neurons, which in turn alters brain plasticity and plays an important role in the reconstruction of sensorimotor circuits through synergistic effects [32–34]. rPMS reduces pain intensity in a variety of disorders, with effectiveness has been demonstrated in clinical studies [35–39]. rTMS modulates neuronal excitability and neural function and has been used globally to treat anxiety, depression, and other psychiatric disorders [40–42]. Some studies suggest that TMS is a reasonable and well-tolerated adjunctive treatment for neuropathic pain, exhibiting long-term analgesic effects on both central and peripheral neuropathic pain [43,44]. CPPS is often treated with a combination of therapies, including psychotherapy, often with significant results [45–49]. TMS of the DLPFC can modulate the emotional value associated with pain, reverse changes in motor cortex excitability caused by pain stimuli, and provide analgesic effects [50]. The application of high-frequency rTMS to the l-DLPFC using a superficial (F8) coil or a deep (H1) coil has been shown to have a distinct and positive impact on alleviating the symptoms of depression [17]. Furthermore, inhibitory rTMS of the r-DLPFC has an anti-anxiety effect [51,52]. Previous research suggested that CPPS is related to central sensitization and requires a multidisciplinary assessment and treatment approach, focusing on improving emotional, physical, and social function to more effectively reduce functional impairment [53]. Some clinical studies have corroborated the hypothesis that the combination of peripheral and central magnetic stimulation is more effective than central magnetic stimulation alone [33,54,55].

The above considerations made obvious the necessity of designing an RCT to study the efficacy and safety of dual-target magnetic stimulation in treating pain and comorbid psychiatric disorders in CPPS. As the above evidence makes it clear, besides causing anatomopathological changes in pelvic floor muscles and impairing sensory processing function, CPPS may also trigger psychological distress manifesting as comorbid anxiety, depression, and reduced QoL. In turn, anxiety and depression can exacerbate the perception of pain, affecting the efficacy of pain treatment. We thus believed it necessary to include in the outcome assessment, in addition to pelvic floor surface electromyography and pudendal nerve evoked potential testing, pain, anxiety, depression, stress, and QoL scales.

### Limitation

In this study we excluded children, adolescents, and individuals aged > 70 years, as to better circumscribe and interpret the study results. However, we considered the age range that meets the standard for marriage and childbirth. Additionally,

sociodemographic factors of the patient population, such as geographical region, lifestyle habits, and healthcare access may also have an impact, indicating potential caveats to be addressed in the design of this single-center trial.

## Conclusions

This study aims to investigate the efficacy and safety of dual-target magnetic stimulation for the treatment of CPPS patients with comorbid psychiatric disorders. Since the latter can deeply influence pain perception and quality of life, identifying psychological barriers is particularly important for healthcare administrators and providers in implementing CPPS intervention strategies as well as in policy development and consensus building. The findings of this study may provide a solid foundation to develop more comprehensive and effective treatment strategies for CPPS.

### Trial status

The experimental protocol has a version 2.0 with a version date of October 20, 2023. The trial is currently in the recruitment phase, which began on January 1, 2024 and will end in December 2025. This study is conducted according to the SPIRIT checklist (S1 File). All patients included in the study were required to complete the randomization form (S2 File).

## Supporting information

**S1 File. SPIRIT checklist.**
(DOCX)

**S2 File. Randomization application form.**
(DOCX)

**S3 File. Trial study protocol (in Chinese).**
(DOCX)

**S4 File. Trial study protocol (in English).**
(DOCX)

## Acknowledgments

The authors would like to thank Professor Yaodong You for his technical assistance.

## Author contributions

**Conceptualization:** Chunmei Luo, Meizhu Zhao.

**Data curation:** Meizhu Zhao.

**Formal analysis:** Jiabei He.

**Funding acquisition:** Chunmei Luo.

**Investigation:** Ren Liu, Lanjin Bai, Xueqian Li, Siyi Tian.

**Methodology:** Haibo Lan, Xiaobin Zhen.

**Resources:** Xiangdong Yang.

**Supervision:** Degui Chang, Xiangdong Yang.

**Writing – original draft:** Chunmei Luo, Jiabei He, Haibo Lan, Meizhu Zhao.

**Writing – review & editing:** Degui Chang, Xiangdong Yang.

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
