## [Decision Letter · Decision Letter 0]

Dear Dr. Yang,

Thank you for submitting your manuscript to PLOS ONE. After careful consideration, we feel that it has merit but does not fully meet PLOS ONE’s publication criteria as it currently stands. Therefore, we invite you to submit a revised version of the manuscript that addresses the points raised during the review process.

We look forward to receiving your revised manuscript.

Kind regards,

Jaber Alizadehgoradel, ph.D

Academic Editor

PLOS ONE

“Chunmei Luo received project funding. This project received support from the Tianfu Scientific Research Incubation Fund (NO. 2022QN02).”

5. Please ensure that you refer to Figure 1 in your text as, if accepted, production will need this reference to link the reader to the figure.

6. We notice that your supplementary figures are uploaded with the file type 'Figure'. Please amend the file type to 'Supporting Information'. Please ensure that each Supporting Information file has a legend listed in the manuscript after the references list.

Reviewers' comments:

Reviewer's Responses to Questions

**Comments to the Author**

1. Does the manuscript provide a valid rationale for the proposed study, with clearly identified and justified research questions?

Reviewer #1: No

Reviewer #2: Yes

2. Is the protocol technically sound and planned in a manner that will lead to a meaningful outcome and allow testing the stated hypotheses?

Reviewer #1: No

Reviewer #2: Yes

3. Is the methodology feasible and described in sufficient detail to allow the work to be replicable?

Reviewer #1: No

Reviewer #2: Yes

4. Have the authors described where all data underlying the findings will be made available when the study is complete?

Reviewer #1: No

Reviewer #2: Yes

5. Is the manuscript presented in an intelligible fashion and written in standard English?

Reviewer #1: No

Reviewer #2: Yes

You may also provide optional suggestions and comments to authors that they might find helpful in planning their study.

Reviewer #1: Important note: This review pertains only to ‘statistical aspects’ of the study and so ‘clinical aspects’ [like medical importance, relevance of the study, ‘clinical significance and implication(s)’ of the whole study, etc.] are to be evaluated [should be assessed] separately/independently. Further please note that any ‘statistical review’ is generally done under the assumption that study specific methodological [as well as execution] issues are perfectly taken care of by the investigator(s). This review is not an exception to that and so does not cover clinical aspects {however, seldom comments are made only if those issues are intimately / scientifically related & intermingle with ‘statistical aspects’ of the study}. Agreed that ‘statistical methods’ are used as just tools here, however, they are vital part of methodology [and so should be given due importance]. I look at the manuscript in/with statistical view point, other reviewer(s) look(s) at it with different angle so that in totality the review is very comprehensive. However, there should be efforts from authors side to improve (may be by taking clues from reviewer’s comments). Therefore, please do not limit the revision only (with respect) to comments made here.

COMMENTS: Unfortunately, there are [as I see/found] many minors as well as few major flaws in the study/manuscript. I have different opinion / observations/concerns or rather questions regarding few issues [including serious ones] which [only vital ones] are given below:

Information given in section ‘Sample size’ (lines 197-202) is not adequate. You assumed the effect size (of 0.5) and quoted an article which is not directly or indirectly(?) related to your topic {according lines 41-43 “This study will aim to assess the efficacy and safety of dual-target magnetic stimulation in CPPS patients with psychological disorders” whereas that article is “Sacral nerve magnetic stimulation combined with extracorporeal shockwave for the treatment of type-ⅢB chronic prostatitis”. Is it correct to use ‘effect size’ from a study in which only the term ‘magnetic stimulation’ is used? Further you said “The pain visual analog score will be used as the primary measurement standard” in which case the sample size has be inflated at least by 10% as the variable is “ordinal”. Software “G*Power” is excellent but the final result [required sample size for this study] will be according to “input”. Since most of the software will yield the result (number of subjects required per arm), I request authors to confirm this [if 54 patients required per arm?].

In this context, I request to kindly note that according to table-2 on page 158 of Jacob Cohen’s paper “A power primer” in Psychological Bulletin, 1992, vol.:112, pp 155-159 [which is a sort of summary of the excellent book by Cohen himself titled ‘Statistical power analysis for the behavioral sciences’, Academic Press, 1977, New York] for medium effect size you need n=64 [for two groups study] or =52 [for three groups study] per group (type-I error=0.05, power=80%). Please note that the ‘effect size’ assumed should have some basis (exact reference needs to be quoted) &/or reasonable/realistic, else the study is very likely ‘not to be able to’ detect a difference despite its presence}.

As stated in the aim {lines 41-43 “This study will aim to assess the efficacy and safety of dual-target magnetic stimulation in CPPS patients with psychological disorders”} that you are assuming existence of some psychological disorders, then why are assessing for presence of psychological disorders {refer to lines 59-60: secondary outcomes will include the Depression, Anxiety and Stress Scale (DASS-21)}. Is not contradictory? In ‘Abstract-Methods’ it is stated that “This prospective, double-blind, randomized controlled trial will recruit 60 CPPS participants and randomly assigned them to three groups (1:1:1) following gender stratification” then in section ‘Study population’ [lines 139-140] it is said that “The study population has no gender restrictions and will include individuals aged 18–70 years, experiencing CPPS accompanied by anxiety or depressions”. Which one is valid or you intend to convey something else? You may be correct, but the question is: How you will execute the recruitment process? [In fact, “How you are executing the recruitment process?” because according section on ‘Trial status’ (lines 403-5) “The trial is currently in the recruitment phase, which began in January 2024 and will end in December 2025. It is still ongoing, and the research is working hard to reach the necessary sample size”.

The difference(s) in [if any] between Repetitive peripheral magnetic stimulation (rPMS) and repetitive transcranial magnetic stimulation (rTMS) has/have not given/brought out/discussed. As stated, “This study will aim to assess the efficacy and safety of dual-target magnetic stimulation in CPPS patients with psychological disorders”, I guess, by “dual-target magnetic stimulation” you mean “rPMS + rTMS. If that is true, the question is ‘Why there are three groups: dual-target magnetic stimulation, peripheral magnetic stimulation, and sham stimulation’? What is this ‘standard treatment’ which all are receiving? That means ‘sham stimulation’ group is ‘active control’ group. Why not work with only two groups: dual-target magnetic stimulation versus/and sham stimulation? What is the logic of having ‘peripheral magnetic stimulation’ group (separate third group)? In short, ‘DESIGN OF THE STUDY’ (one of the very vital/important issue) is not very clear.

In the ‘ABSTRACT’ (lines 41-43) it is stated that “. This study will aim to assess the efficacy and safety of dual-target magnetic stimulation in CPPS patients with psychological disorders” then why you state in lines 107-8:: The aim of this study is to design a high-quality RCT with the stratified randomization of participants? . Question is then ‘which is correct?’ In lines 294-6 you said “A third researcher will …. and distribute the envelopes to the participating patients”. The question is ‘why distribute the envelopes to the participating patients? Is what is described under section “Blinding” [lines 285-308] is really a BLINDING procedure? In fact, at many places section heading is different than the contents [example, section “Randomization and allocation concealment” (lines 213-219)]. In ‘Abstract-Methods’ (lines 44-46) it is stated that “This prospective, double-blind, randomized controlled trial will recruit 60 CPPS participants and randomly assigned them to three groups (1:1:1) following gender stratification“ but the term ‘gender stratification’ does/did not appear anywhere else. Similarly, the term ‘stratified randomization’ was used only in line 107. Regrettably, it seems that ‘gender stratification was forgotten then [as it never reflected later].

Surprising (highly regret to note that) one important term “CONSORT” is not mentioned anywhere in this manuscript. It is well-known that while reporting {findings from and even planning [let the intervention techniques be therapeutic agents, devices, regimes or procedures (i.e. therapeutic trials), preventive ones like vaccines (i.e. prophylactic trials) or rehabilitative, educational etc]}] ‘Clinical Trial’ one should follow CONSORT guidelines. Even important items {like Random Sequence generation (Item 8a), Allocation concealment (Item 9), Blinding (Item 11a)} of/in CONSORT checklist are not described properly [since your article type is ‘Clinical Trial’, you are supposed to cover these items in the report or even in ‘Protocol’ (even if you may not use them) correctly/adquately].

Very good that ‘SPIRIT’ Checklist’ is given in supplementary information file (as S2). However, the accuracy/correctness of the page numbers indicated to content that info seems to be doubtful [example: for item 14 on ‘Sample Size’ you indicated/referred to page 8 but any relevant info is not found there]. Similarly, note that Figure 2: Clinical trial flow chart, similar diagram of flow of sample size is suggested in CONSORT guidelines {available for free download on WWW/NET}.

In “Intervention” section (line 220 onwards) it is stated that “Clinical intervention measures and steps. 1) Before treatment: conduct three checks and seven comparisons…..” and subsequently many things are described. Question is: ‘Are they comparisons (as stated) ‘? If not what they are? Please clarify. Kindly check for the ‘English’ language {this is only as an example; the entire manuscript is full of such examples. One such is in lines 404-5 (the research is working hard to reach the necessary sample size)}. Agreed that English is not our mother tongue (definitely not mine, may or may not be yours but certainly not of many readers), however in any case, remember/ Kindly mind you [please excuse me for such a harsh comment/statement] that this is a scientific/academic document and so all details should be clearly/correctly communicated (do not take reader’s for granted). You may take help of language professional expert, if needed.

Section on ‘Statistical analyses’ (lines 370-76) is extremely ‘inadequate’ (utterly failed to mention even most important/vital techniques). No test is proposed for between groups comparison. Moreover, you stated that “Continuous data will be presented as mean ± standard deviation” but note that all continuous appearing variables may not have/achieved a ‘ratio’ level of measurement. Then Paired t-tests will not be applicable. In this context, I request authors to read the following note pasted from one standard textbook on ‘Medical Research Methodology’ [though I am sure that the authors already know these things].

Though few variables are continuous in appearance they are likely to yield data that are in [at the most] ‘ordinal’ level of measurement [and not in ratio level of measurement for sure]. Then application of suitable non-parametric (or distribution free) test(s) is/are indicated/advisable [even if distribution may be ‘Gaussian’ (also called ‘normal’)]. Agreed that there is/are no non-parametric test(s)/technique(s) available to be used as alternative in all situation(s), but should be used whenever/wherever they are available. Therefore, in short use suitable non-parametric test(s)/technique(s) while dealing with data that are in ‘ordinal’ level of measurement even if [despite that] the distribution may be ‘Gaussian’.

Note that though the measures/tools used are appropriate often times [examples: Pelvic pain score sheet (for females), NIH-CPSI (for males), Anxiety and depression score (DASS-21), SF-36 QoL scale], most of them are likely to yield data that are in ‘ordinal’ level of measurement.

Look at the ‘Conclusions’ [To our knowledge, there have been no literature reports on randomized controlled trials evaluating the impact of dual-target magnetic stimulation on CPPS with comorbid psychiatric disorders. It may serve as a valuable reference for future clinical interventions and research on CPPS with comorbid psychiatric disorders.]. Are they conclusions of/from this study? Basically, ‘are you supposed to draw any ‘Conclusions’ in protocol?’

Moreover, limitations (if any) of the study are not mentioned/listed anywhere. Does that mean {according to authors} there are none? As pointed out in ‘important note’ above “This review pertains only to ‘statistical aspects’ of the study and so ‘clinical aspects’ should be assessed separately/independently [one should carefully consider/look at the clinical implications of the study]. In my opinion, to rescue this article (which seems quite difficult, if not impossible), large amount of re-vision (re-drafting) may be needed. However, please do not limit the revision only (with respect) to comments made here. More improvement is expected. Nevertheless, ‘how to handle/accommodate these suggestions is not questionable as the study is in protocol stage. The respected ‘Editor’ may consider accepting/further processing only if found ‘clinical implications’ valuable [i.e., add(s) to clinical knowledge or positively influence clinical practice]. ‘Major revision’ is recommended [in lieu of plain rejection, to give chance to authors for improvement of the manuscript].

Reviewer #2: Dear authors,

Thank you in advance for the opportunity to review this interesting RCT protocol.

This protocol refers to a future RCT, in recruiting phase, with high scientific value and with potencional high impacto in health. In fact, CPPS consists in an important clinical syndrome with high impact in the QoL of patients.

I give you my congratulations for the idea! I really think this area needs to be investigated!

In this phase, I have two big comments that I will like to be addressed to guarantee my positive opinion to publication:

1) In the diagnostic criteria for generalized anxiety disorder and depressive disorder you used the DSM-5 criteria. Actually there are the new DSM-5-TR criteria that I prefer you to incorporate alongside with the CID-11 criteria for the same disorders. Please, make this actualization.

2) There are an interesting and actual reference that you could incorporate in your introduction and/or discussion: https://doi.org/10.3390/healthcare12050555. Please consider the incorporation.

Thank you!

See you soon!

Good work!

**Do you want your identity to be public for this peer review?** For information about this choice, including consent withdrawal, please see our Privacy Policy

Reviewer #1: No

Reviewer #2: **Yes: ** Bruno Daniel Carneiro

---

## [Author Response · Author response to Decision Letter 1]

8 Feb 2025

Dear Editors and Reviewers:

Thank you for your letter and for the reviewers’ comments concerning our manuscript entitled “Peripheral combined central dual-target magnetic stimulation for rehabilitation of chronic pelvic pain syndrome with psychosomatic disorders: Study protocol for a randomized controlled trial”(ID: PONE-D-24-31808R1). Those comments are all valuable and very helpful for revising and improving our paper, as well as the important guiding significance to our researches. We have studied comments carefully and have made correction which we hope meet with approval.

Revised portion are marked in yellow in the paper. The main corrections in the paper and the responds to the reviewer’s comments are as flowing:

Reviewer #1: Unfortunately, there are [as I see/found] many minors as well as few major flaws in the study/manuscript. I have different opinion / observations/concerns or rather questions regarding few issues [including serious ones] which [only vital ones] are given below:

Response to comment: We feel great thanks for your professional review work on our article. As you are concerned, there are several problems that need to be addressed. According to your nice suggestions, we have made extensive corrections to our previous draft, the detailed corrections are listed below.

Reviewer #1: Information given in section ‘Sample size’ (lines 197-202) is not adequate. You assumed the effect size (of 0.5) and quoted an article which is not directly or indirectly(?) related to your topic {according lines 41-43 “This study will aim to assess the efficacy and safety of dual-target magnetic stimulation in CPPS patients with psychological disorders” whereas that article is “Sacral nerve magnetic stimulation combined with extracorporeal shockwave for the treatment of type-ⅢB chronic prostatitis”. Is it correct to use ‘effect size’ from a study in which only the term ‘magnetic stimulation’ is used? Further you said “The pain visual analog score will be used as the primary measurement standard” in which case the sample size has be inflated at least by 10% as the variable is “ordinal”. Software “G*Power” is excellent but the final result [required sample size for this study] will be according to “input”. Since most of the software will yield the result (number of subjects required per arm), I request authors to confirm this [if 54 patients required per arm?].

In this context, I request to kindly note that according to table-2 on page 158 of Jacob Cohen’s paper “A power primer” in Psychological Bulletin, 1992, vol.:112, pp 155-159 [which is a sort of summary of the excellent book by Cohen himself titled ‘Statistical power analysis for the behavioral sciences’, Academic Press, 1977, New York] for medium effect size you need n=64 [for two groups study] or =52 [for three groups study] per group (type-I error=0.05, power=80%). Please note that the ‘effect size’ assumed should have some basis (exact reference needs to be quoted) &/or reasonable/realistic, else the study is very likely ‘not to be able to’ detect a difference despite its presence}.

Response to comment: We have made correction according to the Reviewer’s comments. This will be the first study to examine the effects of repetitive transcranial magnetic stimulation combined with sacral nerve magnetic stimulation on pain in pelvic pain syndromes, and we used “effect of high-frequency repetitive transcranial magnetic stimulation under different intensities upon rehabilitation of chronic pelvic pain syndrome: protocol for a randomized controlled trial” and “Repetitive Transcranial Magnetic Stimulation: Influence on Stress and Early Responsiveness Outcomes for Depression, Anxiety, and Stress” studies to predict the effect size of this study. We used G*Power software to calculate the total sample size for the study and increased the sample size by 20% based on reviewer comments, as detailed in the highlighted section on lines 227-234.

Reviewer #1: As stated in the aim {lines 41-43 “This study will aim to assess the efficacy and safety of dual-target magnetic stimulation in CPPS patients with psychological disorders”} that you are assuming existence of some psychological disorders, then why are assessing for presence of psychological disorders {refer to lines 59-60: secondary outcomes will include the Depression, Anxiety and Stress Scale (DASS-21)}. Is not contradictory?

Response to comment: Chronic pain is causally related to anxiety and depression, and chronic pain leads to negative emotions such as anxiety and depression . Similarly, anxiety and depression significantly affect the subjective pain experience and QoL, both in the physical and psychological domains, and are closely related to anxiety and depression symptoms in rural Chinese older adults. The efficacy of peripheral magnetic stimulation has been validated in numerous clinical studies, and several studies have shown that peripheral magnetic stimulation is effective in reducing chronic pain. rTMS has been approved by several regulatory agencies for the treatment of TRD, depression comorbid with anxiety, obsessive-compulsive disorder, and other disorders. Currently, some studies have shown that peripheral magnetic stimulation can lead to a reduction in pelvic floor pain in patients with chronic prostatitis, as well as an improvement in patients' depression and anxiety. The aim of our study was to explore the changes in psychological disorders in patients after pain reduction and the effect of modulating emotions on pain.

Reviewer #1: In ‘Abstract-Methods’ it is stated that “This prospective, double-blind, randomized controlled trial will recruit 60 CPPS participants and randomly assigned them to three groups (1:1:1) following gender stratification” then in section ‘Study population’ [lines 139-140] it is said that “The study population has no gender restrictions and will include individuals aged 18–70 years, experiencing CPPS accompanied by anxiety or depressions”. Which one is valid or you intend to convey something else? You may be correct, but the question is: How you will execute the recruitment process? [In fact, “How you are executing the recruitment process?” because according section on ‘Trial status’ (lines 403-5) “The trial is currently in the recruitment phase, which began in January 2024 and will end in December 2025. It is still ongoing, and the research is working hard to reach the necessary sample size”.

Response to comment: According to the reviewers’ comments, we have made extensive modifications to our manuscript. In this revised version, changes to our manuscript are all highlighted in yellow within the document. The revisions in this section can be found in lines 118 - 123 and lines 131 - 140.

Reviewer #1: The difference(s) in [if any] between Repetitive peripheral magnetic stimulation (rPMS) and repetitive transcranial magnetic stimulation (rTMS) has/have not given/brought out/discussed. As stated, “This study will aim to assess the efficacy and safety of dual-target magnetic stimulation in CPPS patients with psychological disorders”, I guess, by “dual-target magnetic stimulation” you mean “rPMS + rTMS. If that is true, the question is ‘Why there are three groups: dual-target magnetic stimulation, peripheral magnetic stimulation, and sham stimulation’? What is this ‘standard treatment’ which all are receiving? That means ‘sham stimulation’ group is ‘active control’ group. Why not work with only two groups: dual-target magnetic stimulation versus/and sham stimulation? What is the logic of having ‘peripheral magnetic stimulation’ group (separate third group)? In short, ‘DESIGN OF THE STUDY’ (one of the very vital/important issue) is not very clear.

Response to comment: We used rTMS to regulate anxiety and depression, while rPMS was used for pelvic floor pain, because pain, anxiety, and depression are causally related to each other, and different treatment protocols may have different effects. rPMS has been validated in many clinical studies, and it plays an important role in the reconstruction of sensorimotor circuits, which can make up for the shortcomings of rTMS. In addition, several clinical studies have also confirmed that rPMS combined with rTMS is better than central magnetic stimulation alone, but whether it is better than peripheral magnetic stimulation alone is yet to be explored, so three groups were set up for a controlled trial. In addition, a number of clinical studies have confirmed that rPMS combined with rTMS is more effective than rTMS alone, but whether the efficacy is better than rPMS alone is yet to be explored, so three groups were set up for a controlled trial. Additionally, in response to your comments, we have added a discussion on the differences between rPMS and rTMS. Please refer to lines 476 - 493.

Reviewer #1: In the ‘ABSTRACT’ (lines 41-43) it is stated that “. This study will aim to assess the efficacy and safety of dual-target magnetic stimulation in CPPS patients with psychological disorders” then why you state in lines 107-8:: The aim of this study is to design a high-quality RCT with the stratified randomization of participants? . Question is then ‘which is correct?’ In lines 294-6 you said “A third researcher will …. and distribute the envelopes to the participating patients”. The question is ‘why distribute the envelopes to the participating patients? Is what is described under section “Blinding” [lines 285-308] is really a BLINDING procedure? In fact, at many places section heading is different than the contents [example, section “Randomization and allocation concealment” (lines 213-219)]. In ‘Abstract-Methods’ (lines 44-46) it is stated that “This prospective, double-blind, randomized controlled trial will recruit 60 CPPS participants and randomly assigned them to three groups (1:1:1) following gender stratification“ but the term ‘gender stratification’ does/did not appear anywhere else. Similarly, the term ‘stratified randomization’ was used only in line 107. Regrettably, it seems that ‘gender stratification was forgotten then [as it never reflected later].

Response to comment: According to the reviewers’ comments, we have made extensive modifications to our manuscript. In this revised version, changes to our manuscript are all highlighted in yellow within the document. The revisions in this section can be found in lines 161 - 162 and lines 242 - 268.

Reviewer #1: Surprising (highly regret to note that) one important term “CONSORT” is not mentioned anywhere in this manuscript. It is well-known that while reporting {findings from and even planning [let the intervention techniques be therapeutic agents, devices, regimes or procedures (i.e. therapeutic trials), preventive ones like vaccines (i.e. prophylactic trials) or rehabilitative, educational etc]}] ‘Clinical Trial’ one should follow CONSORT guidelines. Even important items {like Random Sequence generation (Item 8a), Allocation concealment (Item 9), Blinding (Item 11a)} of/in CONSORT checklist are not described properly [since your article type is ‘Clinical Trial’, you are supposed to cover these items in the report or even in ‘Protocol’ (even if you may not use them) correctly/adquately].

Response to comment: We have meticulously and comprehensively revised the manuscript in strict accordance with the reviewers' invaluable comments. In the current revised version, all the modifications made to the manuscript are conspicuously highlighted in yellow for easy identification.

Our research adheres closely to the CONSORT guidelines. Specifically, for details regarding random sequence generation, please refer to lines 246 - 257; for allocation hiding, lines 259 - 263; and for blinding, lines 263 - 272. The relevant description about following the CONSORT guidelines can be found on lines 146 - 147.

Reviewer #1: Very good that ‘SPIRIT’ Checklist’ is given in supplementary information file (as S2). However, the accuracy/correctness of the page numbers indicated to content that info seems to be doubtful [example: for item 14 on ‘Sample Size’ you indicated/referred to page 8 but any relevant info is not found there]. Similarly, note that Figure 2: Clinical trial flow chart, similar diagram of flow of sample size is suggested in CONSORT guidelines {available for free download on WWW/NET}.

Response to comment: According to your suggestion, we have promptly corrected the page numbers and clinical trial flowchart displayed in the SPIRIT list. We have double-checked to ensure the accuracy of these modifications.

Reviewer #1: In “Intervention” section (line 220 onwards) it is stated that “Clinical intervention measures and steps. 1) Before treatment: conduct three checks and seven comparisons…..” and subsequently many things are described. Question is: ‘Are they comparisons (as stated) ‘? If not what they are? Please clarify. Kindly check for the ‘English’ language {this is only as an example; the entire manuscript is full of such examples. One such is in lines 404-5 (the research is working hard to reach the necessary sample size)}. Agreed that English is not our mother tongue (definitely not mine, may or may not be yours but certainly not of many readers), however in any case, remember/ Kindly mind you [please excuse me for such a harsh comment/statement] that this is a scientific/academic document and so all details should be clearly/correctly communicated (do not take reader’s for granted). You may take help of language professional expert, if needed.

Response to comment: Thanks for your suggestion. We feel sorry for our poor writings, however, we do invite a friend of us who is a native English speaker to help polish our article. And we hope the revised manuscript could be acceptable for you.

Reviewer #1: Section on ‘Statistical analyses’ (lines 370-76) is extremely ‘inadequate’ (utterly failed to mention even most important/vital techniques). No test is proposed for between groups comparison. Moreover, you stated that “Continuous data will be presented as mean ± standard deviation” but note that all continuous appearing variables may not have/achieved a ‘ratio’ level of measurement. Then Paired t-tests will not be applicable. In this context, I request authors to read the following note pasted from one standard textbook on ‘Medical Research Methodology’ [though I am sure that the authors already know these things].

Though few variables are continuous in appearance they are likely to yield data that are in [at the most] ‘ordinal’ level of measurement [and not in ratio level of measurement for sure]. Then application of suitable non-parametric (or distribution free) test(s) is/are indicated/advisable [even if distribution may be ‘Gaussian’ (also called ‘normal’)]. Agreed that there is/are no non-parametric test(s)/technique(s) available to be used as alternative in all situation(s), but should be used whenever/wherever they are available. Therefore, in short use suitable non-parametric test(s)/technique(s) while dealing with data that are in ‘ordinal’ level of measurement even if [despite that] the distribution may be ‘Gaussian’.

Note that though the measures/tools used are appropriate often times [examples: Pelvic pain score sheet (for females), NIH-CPSI (for males), Anxiety and depression score (DASS-21), SF-36 QoL scale], most of them are likely to yield data that are in ‘ordinal’ level of measurement.

Response to comment: Thank you for your valuable feedback on our manuscript.We have revised the Methods section to provide a moredetailed description of the data analysis methods. In this revised version, changes to our manuscript are all highlighted in yellow within the document. The revisions in this section can be found in lines 412-428.

Reviewer #1: Look at the ‘Conclusions’ [To our knowledge, there have been no literature reports on randomized controlled trials evaluating the impact of dual-target magnetic stimulation on CPPS with comorbid psychiatric disorders. It may serve as a valuable reference for future clinical interventions and research on CPPS with comorbid psychiatric disorders.]. Are they conclusions of/from this study? Basically, ‘are you supposed to draw any ‘Conclusions’ in protocol?’

Response to comment: We sincerely thank the editor and all reviewers for their valuable comments, which we used to improve th

---

## [Decision Letter · Decision Letter 1]

Dear Dr. Yang,

Thank you for submitting your manuscript to PLOS ONE. After careful consideration, we feel that it has merit but does not fully meet PLOS ONE’s publication criteria as it currently stands. Therefore, we invite you to submit a revised version of the manuscript that addresses the points raised during the review process.

We look forward to receiving your revised manuscript.

Kind regards,

Jaber Alizadehgoradel, ph.D

Academic Editor

PLOS ONE

Journal Requirements:

Reviewers' comments:

Reviewer's Responses to Questions

**Comments to the Author**

1. Does the manuscript provide a valid rationale for the proposed study, with clearly identified and justified research questions?

Reviewer #1: Partly

Reviewer #2: Yes

2. Is the protocol technically sound and planned in a manner that will lead to a meaningful outcome and allow testing the stated hypotheses?

Reviewer #1: Partly

Reviewer #2: Yes

3. Is the methodology feasible and described in sufficient detail to allow the work to be replicable?

Reviewer #1: No

Reviewer #2: Yes

4. Have the authors described where all data underlying the findings will be made available when the study is complete?

Reviewer #1: Yes

Reviewer #2: Yes

5. Is the manuscript presented in an intelligible fashion and written in standard English?

Reviewer #1: No

Reviewer #2: Yes

You may also provide optional suggestions and comments to authors that they might find helpful in planning their study.

Reviewer #1: COMMENTS: I am happy to see that all most all the comments made on earlier draft are/were addressed/answered. Few are even considered positively & are attended {however not all reasons/arguments are very convincing – though the answers are very comprehensive many are not to the point / focused]. Nevertheless, I recommend the acceptance (yet the editor can decide ‘not to accept’ as original manuscript level/quality is difficult to improve a lot especially if the study is already started which is the case here) only because for revised version much hard work is done by the authors. As a result, the manuscript in present form is ready to publish [with few quality limitations ofcourse], however, following point(s) is/are to be noted and please be incorporated.

According to revised manuscript (lines 122-3: Recruitment will continue until 33 male and 33 female recruits have been registered.) and in lines 119-120 you stated that “Eligible patients, upon providing informed consent, will be categorized by gender and subsequently randomized in a 1:1:1 ratio into the following three groups” is little confusing. Process/method/procedure to be adapted for ‘randomized allocation’ need(s) to be clear. Sample size in each group is 20 i.e. total is 60. But according to lines 122-123 total sample size is 66 [33 male and 33 female recruits]. What is it? How you have achieved (or will achieve) the same sample size [20 or 22 in each group] without using ‘Permuted Block Randomization’ technique? {because there is no such mention}.

Assuming that sample size fixed as 66, you need to give details about how ‘you are going ensure that 11 males and 11 females in each group’ are allocated. In my opinion, sample size fixed at 24 in each group [therefore total sample size is fixed as 72] is more suitable as 24 is completely divisible by randomly chosen permuted block size {3,6} and 2 {M, F}. But even in this case, ‘allocation’ is complicated and can this be called as purely random (or sort of ‘quota’)?

Reviewer #2: The authors addressed all my concerns and I have no more comments.

Good work.

Thank you once more.

**Do you want your identity to be public for this peer review?** For information about this choice, including consent withdrawal, please see our Privacy Policy

Reviewer #1: No

Reviewer #2: **Yes: ** Bruno Daniel Carneiro

---

## [Author Response · Author response to Decision Letter 2]

13 Apr 2025

Dear Editors and Reviewers:

Thank you for your letter and for the reviewers’ comments concerning our manuscript entitled “Peripheral combined central dual-target magnetic stimulation for rehabilitation of chronic pelvic pain syndrome with psychosomatic disorders: Study protocol for a randomized controlled trial”(ID: PONE-D-24-31808R1). Those comments are all valuable and very helpful for revising and improving our paper, as well as the important guiding significance to our researches. We have studied comments carefully and have made correction which we hope meet with approval.

Previously revised parts are still marked in yellow in the thesis, and new revised parts are standardized in green. The main corrections in the paper and the responds to the reviewer’s comments are as flowing:

Reviewer #1: I am happy to see that all most all the comments made on earlier draft are/were addressed/answered. Few are even considered positively & are attended {however not all reasons/arguments are very convincing – though the answers are very comprehensive many are not to the point / focused]. Nevertheless, I recommend the acceptance (yet the editor can decide ‘not to accept’ as original manuscript level/quality is difficult to improve a lot especially if the study is already started which is the case here) only because for revised version much hard work is done by the authors. As a result, the manuscript in present form is ready to publish [with few quality limitations ofcourse], however, following point(s) is/are to be noted and please be incorporated.

Response to comment: Thank you again for your positive comments and valuable suggestions to improve the quality of our manuscript.

Reviewer #1: According to revised manuscript (lines 122-3: Recruitment will continue until 33 male and 33 female recruits have been registered.) and in lines 119-120 you stated that “Eligible patients, upon providing informed consent, will be categorized by gender and subsequently randomized in a 1:1:1 ratio into the following three groups” is little confusing. Process/method/procedure to be adapted for ‘randomized allocation’ need(s) to be clear. Sample size in each group is 20 i.e. total is 60. But according to lines 122-123 total sample size is 66 [33 male and 33 female recruits]. What is it? How you have achieved (or will achieve) the same sample size [20 or 22 in each group] without using ‘Permuted Block Randomization’ technique? {because there is no such mention}.

Assuming that sample size fixed as 66, you need to give details about how ‘you are going ensure that 11 males and 11 females in each group’ are allocated. In my opinion, sample size fixed at 24 in each group [therefore total sample size is fixed as 72] is more suitable as 24 is completely divisible by randomly chosen permuted block size {3,6} and 2 {M, F}. But even in this case, ‘allocation’ is complicated and can this be called as purely random (or sort of ‘quota’)?

Response to comment: Thank you for your comments on our article. According to your suggestions, for inclusion in the study, we used stratification by gender followed by block randomization of participants into 3 groups on a 1:1:1 basis. The description of the modifications has been marked in green in the text and is detailed in lines 65-76 and 120-123. In addition, we also changed the flowchart to have 22 participants in each group.

We appreciate for Editors/Reviewers’ warm work earnestly, and hope the correction will meet with approval. Once again, thank you very much for your comments and suggestions.

Best regards,

Sincerely

Xiangdong Yang

Department of Anorectal, Chengdu Anorectal Hospital, Chengdu, Sichuan Province, China.

Add: 152 Daqiang East Street, Taisheng South Road, Qingyang District, Chengdu City, Sichuan Province, China

Email: xiangdongyang2011@163.com

---

## [Decision Letter · Decision Letter 2]

Dear Dr. Yang,

Thank you for submitting your manuscript to PLOS ONE. After careful consideration, we feel that it has merit but does not fully meet PLOS ONE’s publication criteria as it currently stands. Therefore, we invite you to submit a revised version of the manuscript that addresses the points raised during the review process.

The reviewer noted that despite addressing previous comments, there are still new errors and issues present in this version. Please thoroughly review the manuscript to identify and correct these problems.

We look forward to receiving your revised manuscript.

Kind regards,

Jaber Alizadehgoradel, ph.D

Academic Editor

PLOS ONE

Reviewers' comments:

Reviewer's Responses to Questions

**Comments to the Author**

1. Does the manuscript provide a valid rationale for the proposed study, with clearly identified and justified research questions?

Reviewer #1: Yes

2. Is the protocol technically sound and planned in a manner that will lead to a meaningful outcome and allow testing the stated hypotheses?

Reviewer #1: Partly

3. Is the methodology feasible and described in sufficient detail to allow the work to be replicable?

Reviewer #1: No

4. Have the authors described where all data underlying the findings will be made available when the study is complete?

Reviewer #1: Yes

5. Is the manuscript presented in an intelligible fashion and written in standard English?

Reviewer #1: No

You may also provide optional suggestions and comments to authors that they might find helpful in planning their study.

Reviewer #1: COMMENTS: Although most of the comments made on earlier draft are/were addressed/answered, I am not really very happy with this manuscript [as there are new errors/mistakes]. I feel the allocation description given is not very convincing – how this will ensure gender balance in each group? Please clarify. There are few more issues about which this reviewer is not very convinced [for example: sample size]. Presentation is lengthy but sentences/material is confusing or in non-standard English.

To rescue this article (which seems quite difficult, if not impossible), large amount of re-vision (re-drafting) may be needed. However, please do not limit the revision only (with respect) to comments made here. More improvement is expected. The respected ‘Editor’ may consider accepting/further processing only if found ‘clinical implications’ valuable [i.e., add(s) to clinical knowledge or positively influence clinical practice]. ‘Major revision’ is recommended [in lieu of plain rejection, assuming that the respected editor would like to give chance to authors for improvement of the manuscript].

**Do you want your identity to be public for this peer review?** For information about this choice, including consent withdrawal, please see our Privacy Policy

Reviewer #1: No

---

## [Author Response · Author response to Decision Letter 3]

30 May 2025

Reviewer #1:

Comment 1: Although most of the comments made on the previous manuscript were addressed/answered, I am not very happy with this manuscript [because of the new mistakes/errors]. I don't find the assignment instructions given very convincing - how do you ensure gender balance in each group? Please clarify.

Comment 2: The reviewer was also not very happy with some other issues [e.g. sample size]. The presentation is lengthy, but sentences/materials are confusing or use non-standard English.

Response:

We sincerely appreciate the reviewer's continued scrutiny and valuable suggestions for improving our manuscript.

Response to Comment 1 (Gender Balance): We addressed gender balance by employing stratified randomization. Based on the known gender incidence ratio (male:female = 2:3) of CPPS with psychosomatic disorders at our center, participants were randomized into the three study groups with stratification by gender. The relevant text has been highlighted in yellow in the revised manuscript (Lines 150, 223-232, 244-245). Additionally, we have updated the study flowchart (Figure 3) to clearly reflect the allocation of 25 participants per group.

Response to Comment 2 (Other Issues & Presentation): We thank the reviewer for highlighting these concerns. We have carefully reviewed the sample size justification and presentation throughout the manuscript. The sample size calculation section has been reviewed for clarity. We have also thoroughly edited the entire manuscript to improve conciseness, sentence structure, clarity, and standard English usage. Non-essential details have been removed where possible.

We sincerely thank the Editors and Reviewers for their time and insightful feedback. We hope that the revisions made in response to the comments are satisfactory. Thank you again for your comments and suggestions.

---

## [Editor Report · Decision Letter 3]

Dual-target peripheral and central magnetic stimulation for rehabilitation of chronic pelvic pain syndrome associated with psychosomatic symptoms: study protocol for a randomized controlled trial

PONE-D-24-31808R3

Dear Dr. Yang,

We’re pleased to inform you that your manuscript has been judged scientifically suitable for publication and will be formally accepted for publication once it meets all outstanding technical requirements.

Kind regards,

Jaber Alizadehgoradel, ph.D

Academic Editor

PLOS ONE
---

## [Editor Report · Acceptance letter]

PONE-D-24-31808R3

PLOS ONE

Dear Dr. Yang,

I'm pleased to inform you that your manuscript has been deemed suitable for publication in PLOS ONE. Congratulations! Your manuscript is now being handed over to our production team.

Kind regards,

on behalf of

Dr. Jaber Alizadehgoradel

Academic Editor

PLOS ONE